# PbTe quantum dots highly packed monolayer fabrication by a spin coating method

**Svetlana Lyssenko** , **Michal Amar**, **Alina Sermiagin** , **Refael Minnes** *

Department of Physical Sciences, Ariel University, Ariel, Israel

* refaelm@ariel.ac.il

**Data Availability Statement:** All relevant data are within the manuscript and its Supporting information files.

**Funding:** The author(s) received no specific funding for this work.

## Abstract

This study investigates the fabrication of large-area, highly-ordered monolayers of PbTe quantum dots (QDs) on $TiO_2$/ITO substrate, using a fast, simple, and repeatable spin-coating technique. For the first time, a real monolayer (a layer with the height of a single QD) covering approximately 3 $cm^2$ was successfully prepared, achieving a root-mean-square roughness (Rq) of 1.37 nm. The research systematically explores key parameters such as QD morphology, concentration, spin-coating conditions, substrate characteristics, wetting properties, and solvent effects to optimize thin film deposition. The findings reveal that the spin-coating method favors the formation of layers with spherical QDs (6–9 nm) over cubical QDs (10–13 nm). The findings highlight the significant influence of solvent evaporation rate, viscosity, and wettability on monolayer quality. Chloroform was identified as the optimal solvent for cubical QDs (~90% coverage), while hexane was more effective for spherical QDs (90%-100% coverage). Beyond monolayers, high-quality bilayers were also fabricated, demonstrating the method's potential for multilayer fabrication. This rapid and efficient method for monolayer and bilayer fabrication marks a significant breakthrough in producing uniform, large-area films, facilitating seamless integration with existing technologies. It offers a scalable and cost-effective solution, opening the door to broader applications in fields that demand high-quality thin film deposition.

## 1 Introduction

QDs have been widely explored and utilized across various fields. These include biosensing [1], photocatalysis [2], field-effect transistors [3], thermoelectric converters [4], lasers [5], photovoltaics [6], solar cells [7], and technologies such as proton exchange fuel cell membranes and water desalination systems [8].

Semiconductor quantum dots (QDs) exhibit unique optical properties due to the quantum confinement effect, which transitions continuous energy states to discrete levels as particle size approaches the Bohr exciton radius [9–11]. This tunability enables QDs to cover a wide solar spectrum, making them ideal for various optoelectronic applications [12–15]. Pb-chalcogenide QDs (PbS, PbSe, PbTe) have been extensively studied for their large Bohr exciton radii—18 nm for PbS, 46 nm for PbSe and PbTe—allowing significant carrier confinement and bandgap regulation by size [16]. Among them, PbTe QDs stand out due to their large dielectric

**Competing interests:** The authors have declared
that no competing interests exist.

constant, low electron-hole mass, and strong quantum confinement effects, which make them
particularly suitable for infrared photodetectors, thermoelectric devices, and solar energy har-
vesting [17, 18]. Furthermore, PbTe QDs exhibit a high potential for multiple exciton genera-
tion (MEG), a process where a single high-energy photon generates multiple charge carriers,
enabling higher energy conversion efficiencies in photovoltaic applications [16]. This unique
property provides a pathway to surpass the Shockley-Queasier efficiency limit for single-junc-
tion solar cells, positioning PbTe QDs as a key material for next-generation photovoltaic tech-
nologies [19, 20]. While Cd-based QDs like CdTe and CdSe are widely studied, their
commercialization is hindered by strict regulations and toxicity concerns, as well as the high
cost of complex core/shell structures required to enhance stability [21, 22]. In contrast, PbTe
QDs offer a combination of unique optoelectronic properties and simpler fabrication pro-
cesses, positioning them as a more versatile and impactful material for next-generation
optoelectronic technologies.

PbTe QDs are predominantly utilized in the form of layers and films rather than in their
colloidal state for most applications [23]. A range of techniques, including spin coating [24],
dip coating [25], drop-casting [26], inkjet printing [27], and spray coating [28], are employed
for thin-film fabrication. Spin coating and dip coating are recognized for their ability to pro-
duce thin, uniform QD solids with tunable thickness [24].

Spin coating, in particular, stands out due to its rapid, cost-effective, and scalable nature,
making it a highly favorable method [24]. This technique also allows for repeated deposition
cycles to achieve the desired film thickness, enabling control in layer-by-layer fabrication
[29, 30]. These attributes make spin coating a powerful tool for the production of QD films
suitable for advanced optoelectronic applications. However, achieving a uniform and highly
ordered thin monolayer remains one of the critical challenges in device fabrication and
manufacturing [24].

Layer fabrication of PbTe QD thin films has been an area of active research, with several
studies demonstrating progress in achieving uniform and functional layers. For instance,
Urban et al. utilized a self-assembly approach to create PbTe QD monolayers, focusing on
ligand engineering to enhance interparticle coupling and charge transport [31]. Their work
demonstrated improved electronic properties in the resulting films, making them suitable for
thermoelectric applications. Z. Han et al., employed the thermal evaporation method to
deposit PbTe layers photoconductive detectors [32]. They found that the surface roughness
of the PbTe layers critically influences the optical coupling and device performance. M Pio-
trowski et al., utilized a layer-by-layer spin-coating method to fabricate nanocrystal films using
10 nm PbTe QDs. A uniform and densely packed PbTe nanocrystal film was achieved only
with a thickness of 50 nm [33]. These advancements demonstrate the importance of precise
control over QD surface chemistry and deposition techniques in layer fabrication. Despite
these achievements, challenges remain in scaling up these methods and ensuring reproducibil-
ity for large-area films, which highlights the need for further optimization and innovation in
this field [24].

QDs are initially synthesized by utilizing long-chain insulating ligands. The long-chained
ligands prevent short-range quantum resonance [14]. The electronic properties of QD solids
can be tuned by modification of QDs' surface chemistry via ligand exchange [34]. After QDs
are embedded into solid films, the capping ligands are exchanged with short conductive
organic linkers, such as thiols [35], primary amines [36], carboxylic acids [37], and halide ions
[38], by solid-state ligand exchange [30]. This process improves the QDs coupling in the films,
resulting in enhanced stability, charge transfer, and electrical conductivity [30].

Static spin coater processes are divided into three main steps: deposition, outflow, and evap-
oration [39]. Control over these steps is critical for high-quality layer fabrication. Utilization of

the spin coater provided us with a method for layer fabrication of different varieties. Usually, a spin coating method is based on the dynamics of Newtonian fluids (constant viscosity independent of the stress); however, the QDs solution is defined as a non-Newtonian fluid (stress-dependent viscosity) [40]. Layer fabrication depends on several factors, such as the substrate, the solvent, and the correlation/interaction between liquid-solid-gas interfaces (wetting). More precisely, these interactions are called "dynamic wetting"–the replacement of the gas phase by the liquid phase on the solid surface [41, 42]. Utilizing the spin coater, the spinning spreads the fluid on the surface; hence, forced and not spontaneous dynamic wetting occurs [40]. Wetting is influenced by surface roughness [43], vapor pressure, surface tension, and more [44]. A combination of these parameters may lead us to the best conditions for thin, highly ordered layer fabrication.

As previously mentioned, spin coating, widely used for layer fabrication, offers significant advantages over other methods, including simplicity, cost-effectiveness, and control over film thickness. However, despite its widespread use, there has been little effort in the literature to systematically optimize spin-coating parameters for different QD sizes. This gap is particularly evident for PbTe QDs, where the method's conditions for achieving uniform, high-quality monolayers across various particle sizes remain unexplored.

In this study, we aim to fill this gap by investigating the spin-coating parameters required to prepare monolayers (with a height of a single QD) of PbTe QDs with sizes ranging from 6 nm to 13 nm. Monolayers, not only reduce the use of toxic materials but also enhance the performance of applications where such layers are utilized [45]. To achieve reproducible, high-quality monolayers, we systematically studied the effects of solvents, QD concentrations, and spin-coating parameters. Additionally, the influence of substrate types was analyzed by comparing ITO and $TiO_2$/ITO substrates to understand their impact on layer formation. Our findings provide a detailed understanding of the conditions required for the scalable and efficient preparation of PbTe QD monolayers, contributing valuable insights to the field and advancing the use of PbTe QDs in various applications.

## 2 Materials and methods

### 2.1 Materials

Tri-n-octyl phosphine (TOP) (90% tech.), Te powder (99.999%), squalane (99%), lead acetate trihydrate (99%), oleic acid (OA) (90% tech), tetra-chloroethylene (TCE) (99%), bromobenzene (99%), n-dodecane (99%), and p-phenylene diamine (PDA) (97+%) were purchased from Holand Moran; methanol (AR), chloroform (AR), toluene (AR), benzene (AR), and anhydrous hexane (95%), were obtained from BioLab; isopropanol (<99.8% tech.) and ethanol (99.5%) were purchased from Romical; indium-titanium oxide (ITO), helmanex III were attained from Osilla; Ti-Nanoxide BL/SC (Solaronix); deionized water (> 18.0 MΩ cm$^{-1}$). All chemicals are used as purchased without further purification.

### 2.2 PbTe QDs synthesis

The synthesis of the PbTe QDs was based on the procedure of J. J. Urban *et al.* [31] and modified according to our lab conditions. The synthesis of 6 nm PbTe QDs was performed in a three-neck round-bottom flask under air-free conditions using standard Schlenk line techniques. The reaction involved 570 mg (1.5 mmol) of Pb acetate trihydrate ($Pb(C_2H_3O_2)_2*3$ ($H_2O$)), 1 ml (3.2 mmol) of OA, 14 ml of squalene, and 6 ml of 0.5 M tellurium-tri-n-octyl phosphine solution (TOP-Te). The 0.5 M TOP-Te solution was prepared in advance in a glove box (due to the high sensitivity of TOP to oxygen). The solution was stirred overnight until all the tellurium powder was completely dissolved, and the solution turned transparent and

**Table 1. Synthetic parameters for the synthesis of the various sizes of PbTe QDs.**

| QDs shape | QDs size (nm) | OA volume (ml) | Squalene volume (ml) | TOP-Te concentration (M) | Growth temperature (˚C) |
|---|---|---|---|---|---|
| Spherical | 6.1 ± 0.7 | 1 | 14 | 0.5 | 155 |
| Spherical | 6.9 ± 0.7 | 3 | 12 | 0.5 | 155 |
| Spherical | 7.2 ± 0.7 | 2 | 13 | 0.5 | 157 |
| Spherical | 8.6 ± 0.7 | 3 | 12 | 0.25 | 162 |
| Spherical | 9.8 ± 0.7 | 3 | 12 | 0.75 | 155 |
| Cubic | 11.6 ± 0.9 | 4 | 11 | 0.75 | 154 |
| Cubic | 13.2 ± 1.2 | 6 | 9 | 0.75 | 165 |

yellow (when the tellurium oxidized, the solution would not be completely transparent). Pb/OA/squalene were mixed and heated to 40˚C under vacuum until the bubbling reduced (~15 minutes). The solution bubbles due to the evaporation of acetic acid and water, which are present in Pb acetate trihydrate salt. The reaction mixture was heated to 100˚C for an additional 45 minutes. The vacuum was replaced with $N_2$ gas, and the mixture was heated to 180˚C. 6 ml of a 0.5 M TOP-Te solution were rapidly injected into the reaction mixture. The reaction temperature dropped to ~155˚C and was held at this temperature (± 2˚C) for 2 minutes. The reaction was quenched in a water bath at room temperature. The mixture was washed with 10 ml of hexane into a 50 ml centrifuge tube. The particles were centrifuged for 5 min at 4000 rpm. Further, 20 ml of ethanol were added, and the tube was centrifuged for 5 min at 4000 rpm. The supernatant was disposed of, and the particles were washed with ethanol and redispersed in 10 ml of hexane for storage.

Various PbTe QD sizes were synthesized by changing the OA volume, while the initial solution volume remained at 15 ml with squalene, and the TOP-Te concentration was changed at a fixed 6 ml volume. $Pb(C_2H_3O_2)_2 \cdot 3(H_2O)$ amount, growth time, and injection temperature remained the same. With lower amounts of OA, the particles exhibit a spherical (cuboctahedral) shape, whereas higher volumes of OA result in cubic QDs without any changes to the other synthetic parameters. All synthetic parameters are provided in Table 1.

## 2.3 Layer fabrication

The ITO glass substrates were used as purchased, applying a cleaning procedure provided by the supplier. The glass type of the substrate is polished soda lime, float glass. Initially, the substrates were sonicated in a hot 2% Helmanex III solution for 5 minutes for critical cleaning, then rinsed twice in hot deionized water. Then, additional sonication for 5 minutes in isopropyl alcohol was performed. Finally, the substrates were rinsed twice in deionized water, dried under $N_2$ gas flow, and stored under vacuum without further cleaning. For the Ti-nanoxide ($TiO_2$) deposition, 50 $\mu$l of Ti-nanoxide solution were drop casted, in a static regime, on the ITO glass substrates and spun, in a spin coater, at 5000 rpm for 30 seconds in an $N_2$ glove box. Subsequently, they were annealed at 550˚C for one hour under 6 L/min air flow in a tube furnace with an acceleration of 5˚C/min. Before depositing the PbTe layer, the surface of the $TiO_2$/ITO glass substrate was cleaned. The cleaning process involved drop-casting 100 microliters of solvent onto the substrate during spinning. This procedure was repeated three times to ensure thorough cleaning of the substrate surface. Immediately after, 80 $\mu$l PbTe QDs, at the required concentration, were deposited on a $TiO_2$/ITO glass substrate at the desired speed for 30 seconds and dried under air for at least 30 min before they were used for any purpose.

Layer deposition was performed at room temperature. All the layer preparation parameters are listed in the S1 Table.

## 2.4 Solid-state ligand exchange

Ligand exchange was performed to replace the long OA capping ligand with the shorter linker, phenylenediamine (PDA). The prepared layer was placed in a Petri dish, and 5 ml of 0.1 M PDA in methanol (MeOH) solution were carefully added using a syringe pump with a 10 µl/min flow rate until PDA solution completely covered all the substrate. Ligand exchange was performed for 5 minutes under ambient conditions. After, the fabricated layer was rinsed with methanol three times to remove the excess PDA from the layer and dried using $N_2$ flow. Without ligand exchange, the layer is not stable and can be easily removed from the surface. Successful ligand exchange is indicated by the strong adhesion of the layer to the substrate, making it impossible to remove. Moreover, ligand exchange leads to structures with densely packed particles in close proximity [46].

## 2.5 Layer-by-layer assembly

Layer-by-layer assembly was performed by the layer preparation using the spin coater with subsequent solid-state ligand exchange several times according to the required number of layers. The process was then repeated for a predetermined number of bilayers.

## 2.6 Instruments

Atomic force microscopy (AFM) was used to measure layers' surface roughness. The measurements were carried out using a Bio FastScan scanning probe microscope (Bruker AXS). All images were obtained by applying soft-tapping mode with a FastScan-B (Bruker) silicon probe (spring constant of 1.8 N/m). The measurements were performed under environmental conditions in an acoustic hood to minimize the vibrational noise. The images were captured in the retrace direction at a scan rate of 1.7 Hz. The resolution of the images was 512 samples/line. For image processing and roughness analysis, we used Nanoscope Analysis software. The "flatting" and "plane-fit" functions were applied to each image. All samples were analyzed using a high-resolution scanning electron microscope (HRSEM) (Ultra-High-Resolution Maia 3, Tescan). The QDs' size and morphology were analyzed using a STEM (bright) detector and an electron beam voltage of 25–30 kV. The samples were prepared by drop casting on a copper grid (ultrathin C Type-A, 400 mesh, Ted Pella). A 2 µL of the PbTe/hexane solution was diluted in 150 µL of hexane. Then, 10 µL of the diluted solution was dropped onto a grid. The excess solution on the grid was removed using filter paper. The ImageJ software was utilized for PbTe-QDs size and size distribution calculations. All the image processing was performed by the threshold function. The layer imaging was conducted on HRSEM using a SE detector in beam deceleration mode with an electron beam voltage of 3–5 kV. The layer coverage percentage was calculated by comparing the area of the covered regions to the total substrate area. HRSEM images were taken to capture the material distribution, and ImageJ software was used to measure both the covered regions and the total area. The coverage percentage was determined as the ratio of the covered area to the total substrate area. Multiple images were analyzed, and the results were averaged for accuracy. Energy-dispersive X-ray spectroscopy (EDS) analysis of the layers was performed on HRSEM and obtained by the Aztec microanalytic system (Oxford Instruments) with an electron beam voltage of 12–15 kV. A focused ion beam (FIB) (Helios 5UC) was used for lamella preparation of the multi-layered QDs on the $TiO_2$/ITO/glass cross-section. The high-resolution transmission electron microscope (HRTEM) (JEM 2100, JEOL) was used for lamella imaging, operating at 200 keV. The images were taken

using a GATAN USC 4000 4x4k camera. X-ray diffraction (XRD) patterns were obtained using a PANalytical X'Pert Pro diffractometer with Cu Kα radiation (λ = 1.54 Å) to analyze the crystal structure of the QDs. A spin coater (Osilla) was utilized for layer preparation. Tube furnace KJ-T1200 (Kejia Furnace Co., Ltd., Henan, China) was used for annealing. Contact angle images were captured using a 2MP 1080P Adjustable Smart WiFi USB Digital Microscope, and the measurements were performed using ImageJ software. A volume of 2 µL of double-distilled water and various PbTe QD solutions (dispersed in hexane, TCE, chloroform, toluene, and benzene) was used to evaluate the contact angle on the $TiO_2$ layer surface.

## 3 Results and discussion

This work was performed using cubic PbTe QDs sized 13.2 nm and 11.6 nm, as well as spherical PbTe QDs ranging from 6.1 nm to 9.8 nm in size. Images of all QDs and their corresponding histograms are provided in the S1 Fig. The crystallite structure of the QDs was analyzed using XRD (S2 Fig). The diffraction patterns matched the rock salt crystal structure with an Fm-3m space group and a lattice constant of a = 6.438 Å (PDF 01-078-1904). Peaks at 2θ angles of 27.5, 39.4, 48.7, 56.9, and 64.4 corresponded to the [200], [220], [222], [400], and [420] crystal planes of PbTe QDs, consistent with previously reported data [47, 48]. The crystallite sizes for cubic and spherical NCs, calculated using Scherrer's equation, from 200 peak, were 12.9 nm and 9.8 nm, respectively, aligning well with the sizes observed in HRSEM images. Notably, the crystallite structure was consistent across different QD shapes and sizes, underscoring the uniformity of the crystal structure regardless of morphology or dimensions.

Fig 1a and 1b illustrate the ITO morphology and the surface roughness, measured by HRSEM and AFM, respectively. The ITO layer thickness is approximately 100 µm (Fig 9), which aligns with the data provided by the supplier. The surface roughness, Rq (root mean square roughness), was found to be 3.77 nm. It is well known that ITO is usually used as the first layer on the glass substrate for photovoltaics and light-emitting devices due to its transparency to the sun's spectra [49]. The measured surface roughness was higher by 50% than the one provided by the supplier. The high surface roughness is an additional obstacle for monolayer fabrication.

Initially, we started our layer preparation directly on the ITO glass substrate. Out of all our diverse QDs, the only successful layer was created with 9.8 nm spherical PbTe QDs dispersed in hexane (Fig 2). Fig 2a and 2b presents HRSEM images of a tightly packed layer of 9.8 nm PbTe QDs. Fig 2c and 2d represent an AFM analysis of the same layer. Fig 2c shows AFM analysis with Rq = 10.3 nm for a high scanning area (100 µm²). The Rq roughness was measured to be 7.4 nm for a 4.2 µm² scanning area (Fig 2d). The roughness of higher area increased, due to the PbTe NCs aggregation that occurs in neglectable areas. Due to the ITO's high surface roughness and the provided AFM pictures, it is easy to conclude that the fabricated layer was not a monolayer. A schematic representation of the PbTe QDs on the ITO glass substrate can be found in Scheme 1(a).

Spherical 6.1 nm and cubic 13.2 nm PbTe QDs were utilized for layer fabrication on ITO. The 6.1 nm NCs were too small and filled the ITO surface cavities instead of creating a continuous monolayer. This phenomenon led to the evolution of uneven ITO islands with prominent peaks on the surface (S3 Fig). While the bigger 13.2 nm NCs demonstrated a stacking effect. Instead of being homogeneously dispersed on the ITO surface, big, agglomerated islands were assembled on the surface (S4 Fig). These results led us to confirm that surface roughness is the primary parameter influencing successful layer fabrication. Additionally, the QDs must possess specific properties, including a narrow size distribution and spherical morphology. The appropriate particle size is also critical, as it directly impacts the packing density

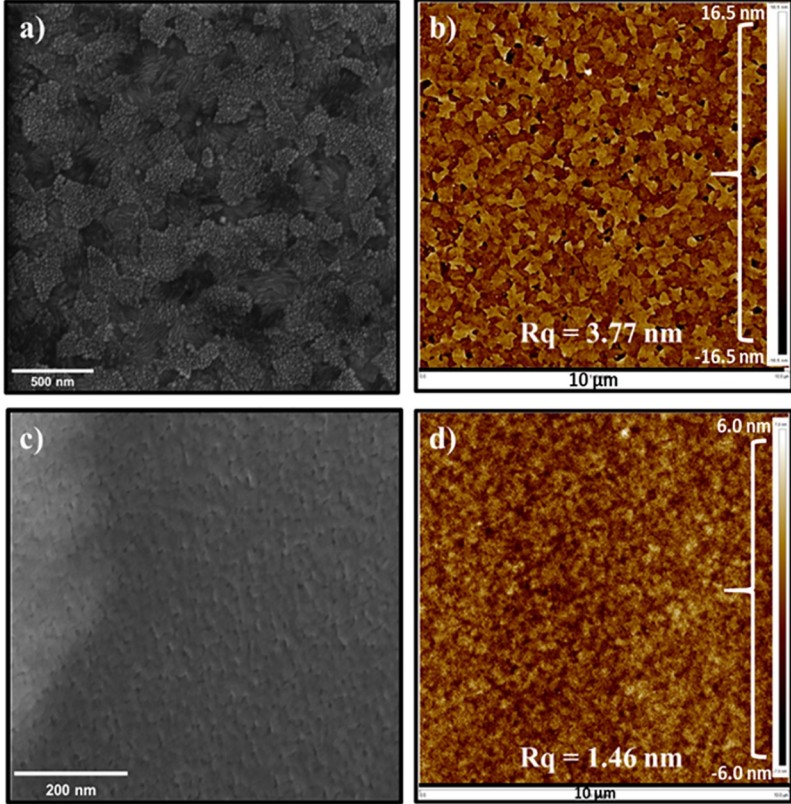

**Fig 1. HRSEM and AFM analyses of ITO glass substrate and TiO$_2$/ITO glass substrate.** HRSEM–with a view filed 2.5 μm (left), and AFM–with a 100 μm$^2$ scan area (right) images of a), b) ITO substrate with Rq = 3.77 nm, c), d) TiO$_2$ monolayer deposited in the ITO substrate with Rq = 1.46 nm.

and uniformity of the layer. The QDs' size has to be large enough to overcome the surface roughness.

To overcome the surface roughness obstacle, a TiO$_2$ layer was deposited on ITO to reduce the surface roughness (Scheme 1b). In addition, TiO$_2$ can act as a hole-blocking layer (from

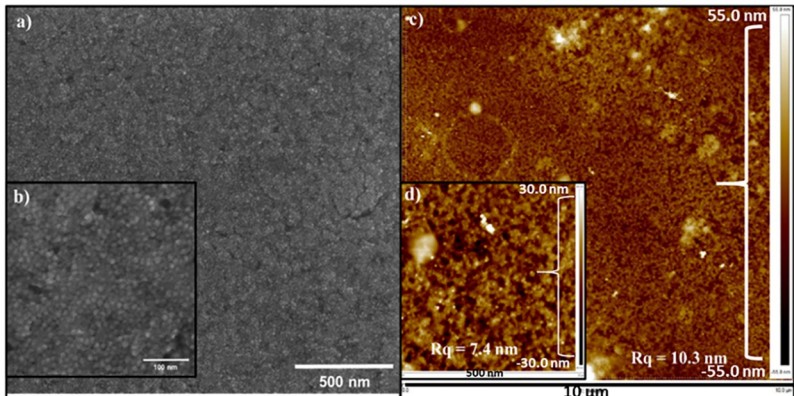

**Fig 2. HRSEM (left) and AFM (right) images of a 9.8 nm PbTe QDs layer prepared on an ITO glass substrate.** a) low magnification with a view field of 2 μm, b) high magnification with a view field of 0.4 μm, c) 100 μm$^2$ scan area with Rq = 10.3 nm, and d) 4.2 μm$^2$ scan area with Rq = 7.4 nm.

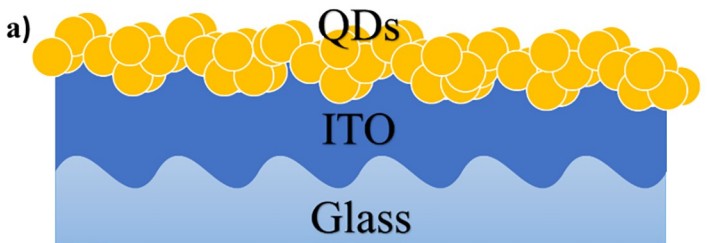

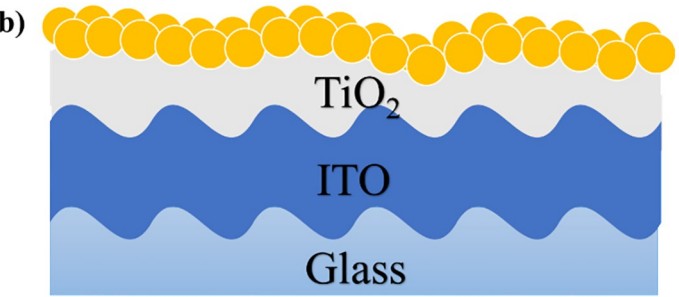

**Scheme 1. A schematic representation of QDs monolayer.** a) QDs monolayer on top ITO glass substrate. b) QDs monolayer $TiO_2$/ITO glass substrate.

electron-hole pairing) and is frequently used in photovoltaic devices [50]. Fig 1c and 1d show HRSEM and AFM images of $TiO_2$ supported on ITO. The $TiO_2$ layer thickness was ~80 μm, as determined from the cross-section image (Fig 9). AFM measurements revealed a substantial 61% reduction in surface roughness, with $TiO_2$/ITO exhibiting an Rq of 1.46 nm compared to 3.77 nm for ITO.

The initial experiment on the $TiO_2$ surface utilized 6.1 nm PbTe QDs. HRSEM and AFM images are depicted in Fig 3. A tightly packed layer over a large area of ~ 3 cm² $TiO_2$/ITO glass substrate, with Rq = 1.37 nm, was successfully prepared. The concentration of NCs in the hexane solvent was 2 mg/ml, the spinning speed was 2000 rpm, and the preparation time was only 30 seconds. To confirm that a tightly fabricated monolayer was achieved across the entire substrate, area mapping using SEM and AFM analyses was performed.

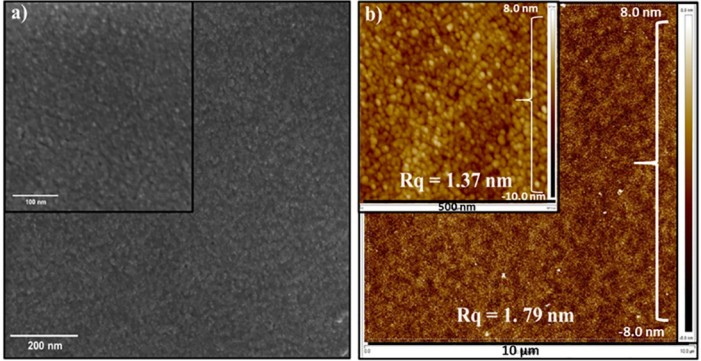

**Fig 3. 6.1 nm PbTe QDs layer on top of $TiO_2$/ITO glass substrate.** a) low magnification HRSEM image with a view field of 1 μm², and high magnification HRSEM image (inset) with a 0.42 μm² view field; b) AFM images of 100 μm² and 0.29 μm² (inset) scan fields with Rq = 1. 79 nm and Rq = 1.37 nm surface roughness, respectively.

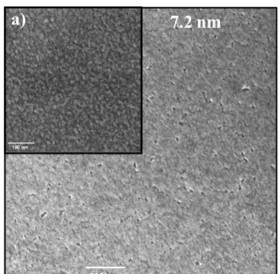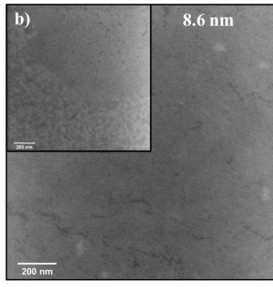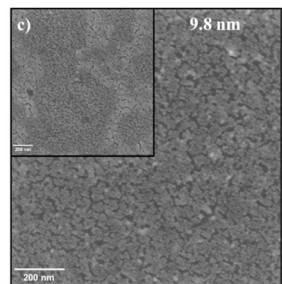

**Fig 4. HRSEM images of PbTe QDs monolayers of various sizes.** a) 7.2 nm, b) 8.6 nm, and c) 9.8 nm, prepared by spin coater on TiO$_2$/ITO glass substrate. Solvent: hexane, spinning speed: 2000 rpm, spinning time: 30 s, QDs concentrations in hexane: 6 mg/ml, 9 mg/ml, and 12 mg/ml for 7.2 nm, 8.6 nm, and 9.8 nm, respectively.

As a result of the successful layer fabrication, we aimed to produce highly packed monolayers from spherical QDs of sizes: 7.2 nm. 8.6 nm, and 9.8 nm (Fig 4a–4c). All the layers were prepared using the spin coater with a spinning speed of 2000 rpm and a spinning time of 30 s, in hexane solvent, at room temperature; the only difference was in the QDs concentration. The concentration of the QD solution was carefully chosen to ensure monolayer formation. A series of experiments were conducted to identify the optimal concentration of each QD size. Fig 4a shows a 7.2 nm PbTe QDs layer on top of the TiO$_2$/ITO glass substrate. On the one hand, a highly packed layer of QDs was successfully fabricated. On the other hand, several holes were found in this layer. These imperfections can influence and reduce the charge transfer through the layer [51]. Additionally, a tightly packed 8.6 nm PbTe QDs monolayer was fabricated on a TiO$_2$/ITO glass substrate. The hexagonal close-packed (hcp) arrays can be detected in Fig 4b. However, some amount of surface cracking was identified in the layer. It is important to mention that a multi-layer can be found in different parts of the measured area.

To improve layer fabrication of this size, another set of experiments was carried out. While maintaining the spin parameters unchanged, QDs concentration in the hexane solution was varied (S5 Fig). A highly ordered layer was prepared from 10 mg/ml PbTe QDs solution. A high number of multi-layered areas was discovered compared to 9 mg/ml (Fig 4b and S5a Fig). This result can be explained by an increase in the QD fluid viscosity. The higher the QD concentration, the higher the fluid viscosity [40]. At lower concentrations (8 mg/ml), a different phenomenon was observed (S5 Fig). The quantity of QDs on the surface was insufficient to fully cover the substrate and form a uniform layer. Under these conditions, the fluid outflow during the spin-coating process became the dominant factor influencing layer formation. Fig 4c shows a non-uniform multilayer composed of 9.8 nm PbTe QDs, characterized by the presence of numerous holes and cracks, indicating a lack of tight packing. The inset in Fig 4c reveals a secondary gradient flow of the QD fluid atop the initial layer, a phenomenon that may be attributed to the Marangoni effect. This effect takes place when a surface tension gradient occurs during the layer assembly progress [52].

One of the main aspirations of this work was the creation of tightly packed large areas of monolayered QDs of different sizes. For this reason, the QD size was increased above 10 nm. It is important to mention that QD morphology changes from spherical to cubical around 10 nm [31]. Initially, cubical QDs, 13.2 nm were deposited over TiO$_2$/ITO glass substrate hoping different surface roughness would provide better results than previously (Fig 5 and S4 Fig). Control over the surface roughness can provide better surface wetting [43]. When the fabricated layer showed underwhelming result, the solvent was changed to improve wetting. The

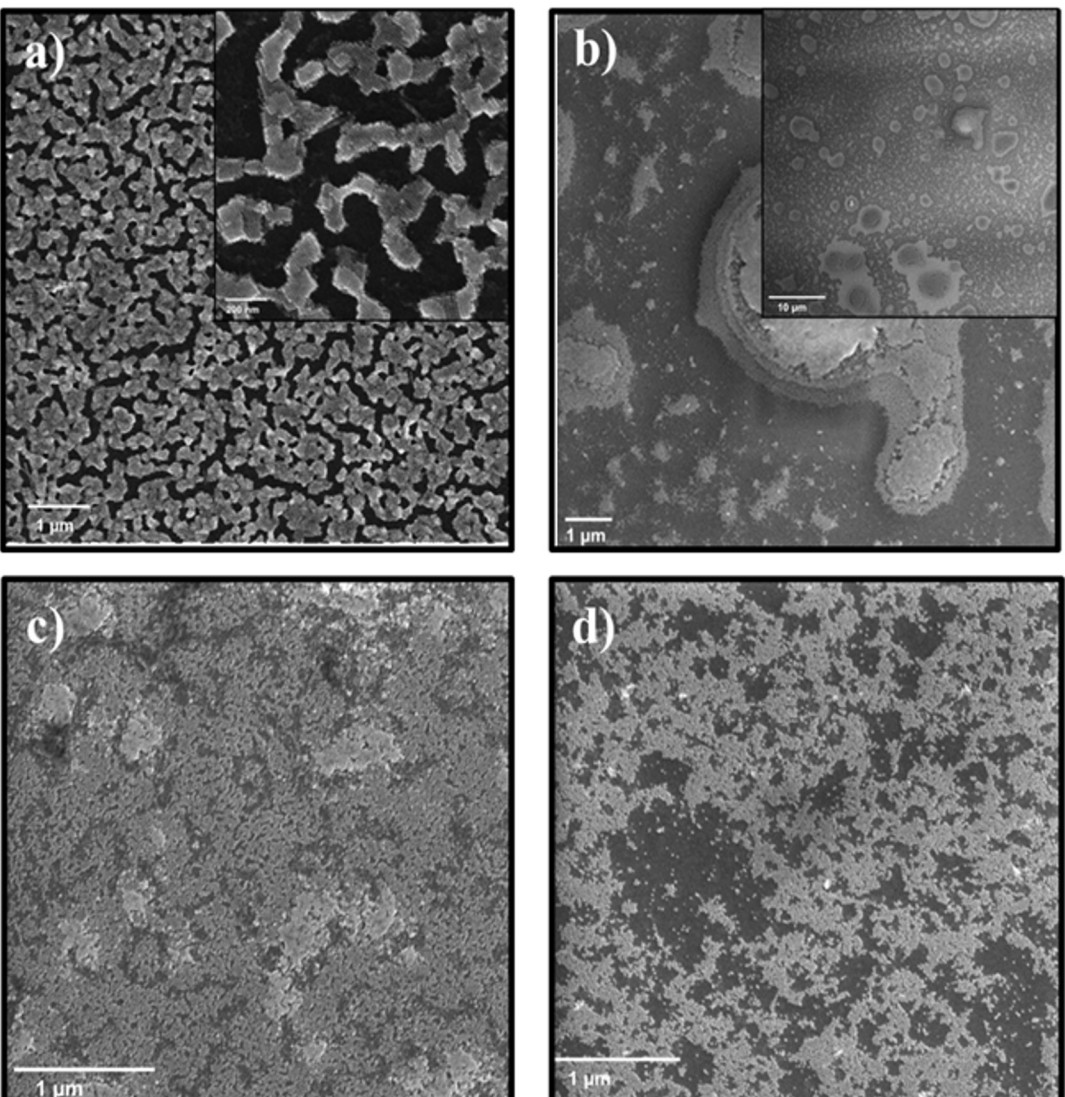

**Fig 5. HRSEM images of 13.2 nm PbTe QDs layers, prepared by a spin coater, using different solvents.** a) hexane, b) toluene, c) CHCl$_3$, and d) TCE. Spinning speed: 3000 rpm, spinning time: 30 s, PbTe concentration in solvents: 20mg/ml.

synthesized QDs were stored in hexane, and for this reason, it was chosen as the main solvent for layer preparation.

The wettability of a surface depends not only on the surface roughness but also on liquid-solid contact parameters [43]. Meaning that the solvent characteristics play a critical role in layer fabrication. To develop an understanding of the layer fabrication process, various solvents were utilized with diverse characteristics, listed in Table 2. Solvents' selection was based on several parameters, including vapor pressure (v.p), evaporation rate (e.r), viscosity (v), surface tension (s.t), and polarity. Polarity, hydrophobicity, and hydrophilicity properties influence the contact angle, hence influencing the whole wetting process [53, 54].

The chosen solvents were TCE, toluene, chloroform, N-dodecane, benzene and bromobenzene. The contact angles of the PbTe QDs dispersed in the mentioned solvents range between 16 and 25 degrees, indicating good wettability on the TiO$_2$ layer (S6 Fig). Despite successful

**Table 2. Various parameters of solvent utilized for layer fabrication [55, 56].**

| Compound | Boiling Point (˚C) | Vapor Pressure at 25˚C (kPa) | Evaporation Rate | Surface Tension (N/cm) | Dynamic Viscosity (mPa*s) |
|---|---|---|---|---|---|
| Butyl acetate | 126 | 1.66 | 1.00 | $25.30*10^{-5}$ (25˚C) | 0.68 (25˚C) |
| Hexane | 69 | 20.1 | 8.10 | $17.91*10^{-5}$ (25˚C) | 0.28 (30˚C) |
| Chloroform | 61.2 | 26.6 | 11.6 | $27.16*10^{-5}$ (25˚C) | 0.53 (30˚C) |
| TCE | 121 | 2.53 | 6.00 | $44.40*10^{-5}$ (25˚C) | 0.84 (25˚C) |
| Toluene | 110 | 2.91 | 2.24 | $28.53*10^{-5}$ (25˚C) | 0.55 (30˚C) |
| Benzene | 80 | 9.95 | 5.10 | $28.90*10^{-5}$ (20˚C) | 0.60 (30˚C) |
| Bromobenzene | 156.2 | 0.55 | *** | $36.50*10^{-5}$ (20˚C) | 1.08 (25˚C) |
| N-dodecane | 215 | 0.02 | *** | $22.29*10^{-5}$ (30˚C) | 0.79 (30˚C) |
| Water | 100 | 3.17 | *** | $72.80*10^{-5}$ (25˚C) | 0.89 (30˚C) |

PbTe QDs dispersion in benzene and bromobenzene only partial formation of the layers was achieved. N-dodecane was found to be too oily, and NCs were sliding off the substrate surface. N-dodecane is known as an oily compound with high viscosity and low vapor pressure. Cubical 13.2 nm QDs fabricated layers are depicted in Fig 5a–5d, and the preparation parameters are disclosed in the S1 Table.

A fabricated layer using hexane as a solvent was a multi-layer with extensive cracking (Fig 5a). A nearly fixed crack width, suggests that the layer cracked during/after the evaporation of the solvent. Moreover, dynamic wetting properties, such as viscosity and solvent evaporation rate, strongly influence layer formation. As was mentioned above, viscosity increases with the increase in the QD concentration. In this instance, the high QD concentration of 20 mg/ml, combined with the slow evaporation rate of hexane, resulted in the formation of a multilayered structure. These factors, along with ligand exchange, contributed to the observed outcome.

In solid-state ligand exchange, long capping ligands are exchanged with shorter alternatives. Usually, solid-state ligand exchange improves the layer's electronic properties [57]. Ligand exchange process is based on two main parameters: chemical bond strength and the steric effect [58, 59]. These parameters dramatically affect layer fabrication. An additional important factor is ligand length, which can maintain and tailor the distances between the QDs on the substrate surface [60]. Based on these facts, it should be mentioned that the ligand exchange process is sometimes responsible for the large volume construction of the deposited film; hence it leads to surface cracking [57].

The suspended QDs in toluene, formed large aggregates over $TiO_2$/ITO glass substrate, resulting in a non-homogeneous layer (Fig 5b). Large agglomerates formed due to the low evaporation rate of the solvent (e.r. = 2.24), which resulted in slower layer formation and a prolonged self-assembly process of the QDs. In chloroform, a predominantly monolayer structure was formed at a relatively high surface coverage, while small bilayer islands were observed (Fig 5c). The fabricated layer with TCE presented low surface coverage, and lone particles on the surface at the same spin parameters and QDs concentrations (Fig 5d).

While comparing various solvents performance at fixed spin coating parameters (Spinning speed: 3000 rpm, spinning time: 30 s) chloroform was found as the best solvent for cubical QDs layer fabrication. Presumably, regards its highest evaporation rate, relatively low dynamic viscosity, and high vapor pressure. However, the layer was not flawless, and additional tuning was needed. A series of experiments with different PbTe QDs concentrations was carried out for additional tuning of the layer (S7 Fig). Furthermore, a variety of spinning speeds from 1500 to 3500 rpm were applied as well (S8 Fig). Best conditions were achieved with 20 mg/ml PbTe QDs in chloroform, 3000 rpm, for 30 seconds (Fig 5c and S7b Fig).

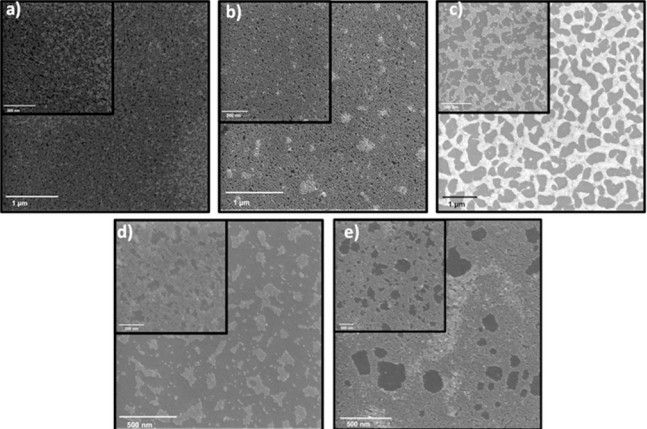

**Fig 6. HRSEM images of 11.6 nm PbTe QDs layers.** Layers prepared by a spin coater, with spinning time of 30 s, PbTe QDs concentration of 11 mg/ml, and spinning speed 3000 of rpm in different solvents: a) hexane, b) $CHCl_3$ c) toluene, d) mix of $CHCl_3$ and hexane in a volumetric ratio of 1:1, and e) mix of toluene and hexane in a volumetric ratio of 1:1.

NC size effect on cubical QDs monolayer fabrication was tested while sizing down the NCs from 13.2 nm to 11.6 nm. Different solvents and various solvent mixtures such as hexane (Fig 6a), chloroform (Fig 6b), toluene (Fig 6c), 1:1 chloroform/hexane (Fig 6d), and 1:1 toluene/ hexane (Fig 6e), were applied to create a monolayer within the stated parameters. The best layers were produced in hexane and chloroform with the best coverage so far ~90%. In chloroform, only a small number of areas were uncovered or double-layered compared to the layers in hexane with ~50% multi-layered surface. Layer fabrication optimal conditions were found to be 11 mg/ml in chloroform at 3000 rpm for 30 s. It is imperative to point out that although the fabricated layer in toluene had a low surface coverage, it showed highly packed islands of monolayers (inset of Fig 6c).

Mixing two solvents with different properties, such as low and high evaporation rates is a known strategy for spin coating conditions optimization. The combination of the solvents leads to two different parallel processes. Fast evaporation of one solvent gives a high surface coverage and uniform film thickness, while slow evaporation of the second solvent allows the NCs self-assembly before the film fabrication process is completed [61–63]. Accordingly, experiments with two different solvent mixtures were conducted. The fabricated layer created in a 1:1 chloroform/hexane mixture was unsuccessful, presumably due to the noncomputability of the solvents caused by their similarity (Fig 6d). On the other hand, the combination of hexane and toluene (1:1) showed promising behavior (Fig 6e).

Further experiments with a toluene/hexane mixture were executed at lower spinning speeds; 1000 rpm and 2000 rpm. Multi-layers were formed at lower spinning speed while at higher spinning speed a packed monolayer with some multi-layered areas was created (Fig 7a and 7b). As long as the spinning speed is low, it maintains a higher number of QDs on the surface. Both layers show higher coverage compared to previous experiments. However, the best monolayer remains in chloroform only. The fact that full surface coverage was achieved only with a spherical shape, but not cubical leads to a conclusion that film fabrication is deeply influenced by NC morphology. All various disclosed fabricated layers and PbTe QD are presented in S1 Table and S1 Fig, respectively.

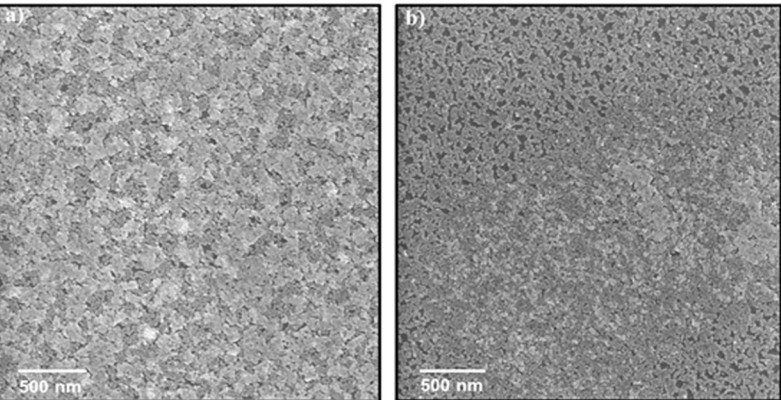

**Fig 7. HRSEM images of 11.6 nm PbTe QDs layers, prepared by a spin coater.** Spinning time of 30 s, PbTe QDs concentration of 11 mg/ml in a mix of toluene and hexane with a volumetric ratio of 1:1, and spinning speed of: a) 1000 rpm, b) 2000 rpm.

A bi-layered structures of various sizes were prepared by a layer-by-layer method. QD multilayers improve charge transfer properties and overall device performance [29]. A bi-layer of 6.1 nm QDs layer on the 7.9 nm QDs layer, 6.9 nm QD layer on the 6.1 nm QD layer, and 7.9 nm QDs layer on the 6.1 nm QDs layer were successfully prepared with high quality (Fig 8). All layers are tightly packed, highly ordered, and exhibit extensive area coverage (view field ~1.5 μm). A cross-section analysis was conducted to measure ITO, $TiO_2$, and QD layer thickness (Fig 9). The bottom layer in the cross-section image is glass, the next layer is ITO, the third layer is $TiO_2$, and the last illustrated layer is the QDs bilayer of 7.9 nm on a 6.1 nm layer. The measured thickness of the quantum dot (QD) layer was 15 nm, matching the expected thickness for the bilayer, which is the combined size of 6.1 nm and 7.9 nm QDs. The EDS analysis of each layer can be found in S9 Fig.

The spin coating technique effectively produces highly packed and ordered monolayers of QDs by controlling parameters such as surface roughness, wettability, polarity, and QD concentration. Our findings highlight the significant role of QD morphology in layer fabrication. Spherical QDs are more adaptable for tuning, whereas cubical QDs present greater challenges, and the spin coating method appears inadequate for their assembly. For larger QDs, higher concentrations, and faster spinning speeds are necessary to achieve proper layer formation.

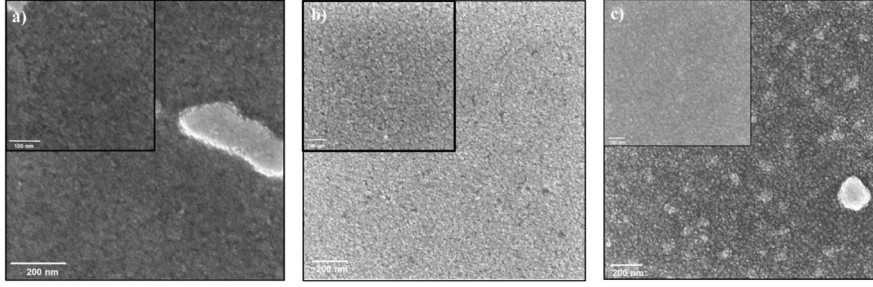

**Fig 8. HRSEM pictures of the layer-by-layer assembly of QDs bi-layers.** a) 6.1 nm NCs layer fabricated on a 7.9 nm NCs layer, b) 6.9 nm NCs layer fabricated on a 6.1 nm NCs layer, and c) 7.9 nm NCs layer fabricated on a 6.1 nm NCs layer, using the spin coater method. Solvent: hexane; spinning speed: 2000 rpm; spinning time: 30 s; NCs concentration in hexane: 2 mg/ml, 6 mg/ml, and 8 mg/ml.

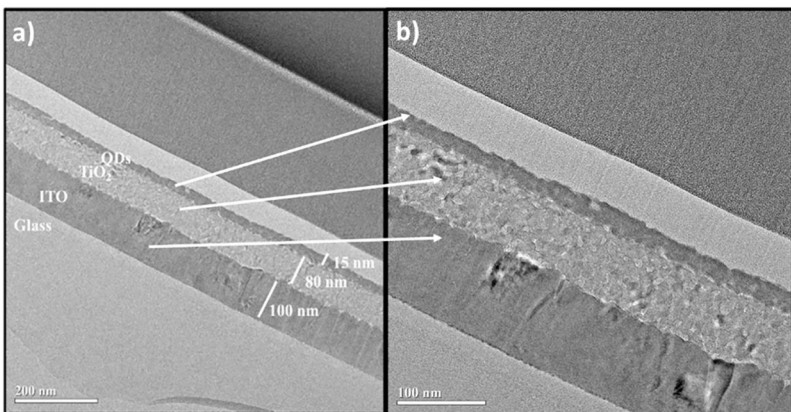

**Fig 9. Cross-section HRTEM images of the multi-layered PbTe QDs on TiO$_2$/ITO glass substrate, prepared by the layer-by-layer method.** a) low magnification, and b) high magnification HRTEM images.

Surface roughness is a critical factor in creating quality layers, as high substrate roughness disrupts QD ordering in nanocrystal (NC) solids. Smoother surfaces yield better results.

Among tested the solvents, hexane, and chloroform proved to be the most effective for layer fabrication, likely due to their favorable wetting properties. As previously discussed, the wetting behavior of a film is influenced by factors such as the viscosity, evaporation rate, and vapor pressure of the QD solution. Higher QD concentrations increase fluid viscosity leading to island and multilayer formations, resulting in surface cracking. Conversely, very low concentrations result in insufficient substrate coverage. PbTe QDs, being hydrophobic, form well-ordered layers on hydrophobic TiO$_2$/ITO glass substrates [64]. A water contact angle of 55˚ on a TiO$_2$ surface, compared to a hexane contact angle of 11˚, confirms that the TiO$_2$/ITO surface is predominantly hydrophobic rather than hydrophilic (S6 Fig) [65–67].

From a series of experiments on layer formation using hexane, chloroform, toluene, and their mixtures, it was concluded that evaporation rate is the most critical solvent parameter influencing layer formation. It was observed that hexane is the optimal solvent for spherical QDs, while chloroform is more suitable for cubical QDs. The relatively low evaporation rate of hexane, compared to chloroform, allows more time for the self-assembly of spherical QDs during the spinning process, whereas the high evaporation rate of chloroform prevents the agglomeration of cubical QDs. Therefore, solvents with low viscosity, high vapor pressure, and high evaporation rates are considered essential for achieving tightly packed, ordered layers using the spin coating method. In Table 3, the surface roughness (Rq) of QD layers prepared by various methods and solvents is compared with previously reported data.

**Table 3. Comparison of various QD layers.**

| QDs | Size (nm) | Preparation Method | Solvent | Substrate | AFM Rq (nm) | Ref. |
|---|---|---|---|---|---|---|
| CdSe | 4.7 | Drop casting | Cyclohexane | ITO | 5.71 | [68] |
| CdSe | 3.3 | Drop casting | Cyclohexane | ITO | 4.52 | [68] |
| CdSe/ZnS | ~ 10 | Spin-coating | Hexane | ITO | 2.5 | [69] |
| PbS | ~ 5 | Spin-coating | Toluene | ZnO/ITO | 9.1 | [70] |
| PbS | ~ 4.5 | Spin-coating | Toluene | ITO | 3.2 | [71] |
| CsPbI$_3$ | ~ 10 | Spin-coating | n-octane | TiO$_2$/FTO | 9.8 | [72] |
| PbSe | ~ 10 | Spin-coating | octane | ZnO/ITO | 2.5 | [73] |

## 4 Conclusions

This study demonstrates that spin-coating is a highly effective method for fabricating monolayers and bilayers of PbTe quantum dots. The experiments highlighted the critical role of substrate surface roughness, QD morphology, and solvent properties in achieving tightly packed and uniform layers. While $TiO_2$/ITO substrates significantly improved layer quality due to their reduced surface roughness, solvent optimization emerged as a key factor in layer formation. Hexane and chloroform were identified as the most effective solvents, offering tunability based on QD morphology. Although spherical QDs produced better results, challenges remain with cubical QDs, requiring further methodological improvements. The findings pave the way for scalable, cost-effective, and reproducible fabrication of QD layers for advanced applications in optoelectronics and photovoltaics. Future research should focus on overcoming the limitations in layer quality for larger, cubical QDs and expanding the applicability of this method to other materials and device architectures.

## Supporting information

**S1 Fig. HRSEM images and the corresponding size distribution histograms of PbTe QDs.** a) 13.2 ± 1.1 nm, b) 11.6 ± 0.9 nm, c) 9.8 ± 0.7nm, d) 8.6 ± 0.6 nm, e) 7.9 ± 0.71 nm f) 7.2 ± 0.7 nm, g) 6.9 ± 0.4 nm, h) 6.1 ± 0.5 nm, used for layer fabrication on aTiO2/ITO glass substrate, by spin coater method.
(TIF)

**S2 Fig. Powder XRD patterns of PbTe QDs and PbTe pattern from the database -PDF 01-078-1904.**
(TIF)

**S3 Fig. SEM images of 6.1 nm PbTe QDs layer on ITO glass substrate.** a) low magnification, b) high magnification, prepared by spin coater. Spinning speed: 2000 rpm, spinning time: 30 s, PbTe QDs concentration in hexane–5 mg/ml.
(TIF)

**S4 Fig. SEM images of 13.2 nm PbTe QDs layer on ITO glass substrate.** Layer prepared by spin coater with—spinning speed: 2000 rpm, spinning time: 30 s, PbTe concentration in hexane–10 mg/ml.
(TIF)

**S5 Fig. SEM images of 8.6 nm PbTe QDs layers at different concentrations.** a) 10 mg/ml, and b) 8 mg/ml prepared by spin coater on $TiO_2$/ITO glass substrate. Solvent: hexane, spinning speed: 2000 rpm, spinning time: 30 s.
(TIF)

**S6 Fig. Contact angle of water and various PbTe QDs solutions on $TiO_2$/ITO glass substrate.**
(TIF)

**S7 Fig. SEM images of 13.2 nm PbTe QDs layers.** Layers prepared by a spin coater, with a spinning speed of 3000 rpm and spinning time of 30 s, using $CHCl_3$ as a solvent, at different concentrations of PbTe QDs. a) 10 mg/ml, b) 20 mg/ml, c) 25 mg/ml, and d) 30 mg/ml.
(TIF)

**S8 Fig. SEM images of 13.2 nm PbTe QDs layers.** Layers were prepared by a spin coater, spinning time of 30 s, PbTe QDs concentration of 25 mg/ml in $CHCl_3$ with different spinning

speeds. a) 1500 rpm, b) 2000 rpm c) 2500 rpm, d) 3000 rpm, and e) 3500 rpm.
(TIF)

**S9 Fig. EDS spectra of different layers.** a) PbTe QDs layer, b) TiO$_2$ layer, and c) ITO layer.
(TIF)

**S1 Table. Experimental list/parameters for layers fabrication at various PbTe QDs sizes.**
(DOCX)

## Acknowledgments

Svetlana Lyssenko wishes to thank Ariel University for the Ph.D student fellowship.

## Author Contributions

**Conceptualization:** Svetlana Lyssenko, Refael Minnes.

**Formal analysis:** Svetlana Lyssenko, Alina Sermiagin.

**Investigation:** Svetlana Lyssenko, Michal Amar.

**Methodology:** Svetlana Lyssenko, Michal Amar, Refael Minnes.

**Resources:** Refael Minnes.

**Supervision:** Refael Minnes.

**Validation:** Svetlana Lyssenko, Alina Sermiagin.

**Visualization:** Svetlana Lyssenko, Alina Sermiagin.

**Writing – original draft:** Svetlana Lyssenko, Alina Sermiagin.

**Writing – review & editing:** Svetlana Lyssenko, Alina Sermiagin, Refael Minnes.

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
