## [Decision Letter · Decision Letter 0]

13 Nov 2024

PONE-D-24-40936PbTe Quantum Dots Highly Packed Monolayer Fabrication by a Spin Coating MethodPLOS ONE

Dear Dr. Lyssenko,

Thank you for submitting your manuscript to PLOS ONE. After careful consideration, we feel that it has merit but does not fully meet PLOS ONE’s publication criteria as it currently stands. Therefore, we invite you to submit a revised version of the manuscript that addresses the points raised during the review process.

After careful evaluation by reviewers, I believe your manuscript has potential, but significant revisions are necessary to enhance its clarity, rigor, and overall contribution.Please address the reviewers comments in your revised manuscript and provide a detailed response to each point raised, explaining how the revisions address the feedback. Once revised, we will reassess your manuscript for further consideration.

We look forward to receiving your revised manuscript.

Kind regards,

María Eugenia Pérez Barthaburu

Academic Editor

PLOS ONE

Journal Requirements:

2. We note that your Data Availability Statement is currently as follows: [All relevant data are within the manuscript and its Supporting Information files.] Please confirm at this time whether or not your submission contains all raw data required to replicate the results of your study. Authors must share the “minimal data set” for their submission. PLOS defines the minimal data set to consist of the data required to replicate all study findings reported in the article, as well as related metadata and methods (https://journals.plos.org/plosone/s/data-availability#loc-minimal-data-set-definition). For example, authors should submit the following data: - The values behind the means, standard deviations and other measures reported; - The values used to build graphs; - The points extracted from images for analysis. Authors do not need to submit their entire data set if only a portion of the data was used in the reported study. If your submission does not contain these data, please either upload them as Supporting Information files or deposit them to a stable, public repository and provide us with the relevant URLs, DOIs, or accession numbers. For a list of recommended repositories, please see https://journals.plos.org/plosone/s/recommended-repositories. If there are ethical or legal restrictions on sharing a de-identified data set, please explain them in detail (e.g., data contain potentially sensitive information, data are owned by a third-party organization, etc.) and who has imposed them (e.g., an ethics committee). Please also provide contact information for a data access committee, ethics committee, or other institutional body to which data requests may be sent. If data are owned by a third party, please indicate how others may request data access.

Reviewers' comments:

Reviewer's Responses to Questions

**Comments to the Author**

1. Is the manuscript technically sound, and do the data support the conclusions?

Reviewer #1: Partly

Reviewer #2: Partly

2. Has the statistical analysis been performed appropriately and rigorously? 

Reviewer #1: N/A

Reviewer #2: N/A

3. Have the authors made all data underlying the findings in their manuscript fully available?

Reviewer #1: Yes

Reviewer #2: Yes

4. Is the manuscript presented in an intelligible fashion and written in standard English?

Reviewer #1: Yes

Reviewer #2: No

5. Review Comments to the Author

Reviewer #1: Ms. Ref. No.: PONE-D-24-40936 PLOS ONE

Title: PbTe Quantum Dots Highly Packed Monolayer Fabrication by a Spin Coating Method

Comments:

This paper describes the synthesis and characterization of large-area ultrathin layer(s) of PbTe quantum dots(QDs) by the spin coating technique. By optimizing different deposition parameters, the authors claimed that they achieved compact and coherent coating of the large area (~ 3 cm2) substrate with mono/bi-layers of PbTe QDs. This work has made some intriguing and important research progress, but before accepting the manuscript for publication, the authors must address/revise the following issues:

1. In Abstract: The author(s) are advised to avoid vague information and add some key results and numerical findings, mention the type/nature of substrates, processing temperature(s) used for readers. The abstract is detailed but somewhat repetitive, particularly when discussing large-area coating of the underlying substrates, concentrations of coating solutions, etc. Consider condensing the text to avoid redundancy, while still conveying the essential findings.

2. Introduction: Please highlights the motivation behind this study, the novelty of your research, and potential contribution to the literature. There are numerous reports in the literature on PbTe QDs.

(a) The author should justify why they focused on PbTe layer(s) over other important materials, for example, CdTe etc., by highlighting its beneficial optoelectronic features for readers. For example, see Ref1.

Ref.1 Optical Materials 155, 115816, 2024. https://doi.org/10.1016/j.optmat.2024.115816

(b) In lines#42-45, the authors mentioned quantum confinement effect and Bohr excitation radius etc. of materials under study. The authors are advised to elaborate the discussion and provide numerical values of Borh excitation radius for readers with solid first-hand ref(s).

3. Experimental section: (a) Give replace the “Instruments” section at the end. (b) Mention clearly how STEM/TEM specimens of QDs were prepared, which solvents were used for QDs dispersion to be drop-casted on Cu-grid. Also mention the grid size/dimensions for readers. (c) The authors mentioned that they annealed TiO2 thin films grown on ITO/glass substrates at 550C at heating rate of 5C/min, while the glass softening temperature is well below 550C, how they avoid the microcrack of coating layers at such high temperature?

4. Results and Discussions.

(a) In lines#185-189, the authors mentioned the different factors that affect the compact and coherent coating of the substrates of interest. For growing nucleation centers, a clean surface of the desired substrates is a very important factor [Ref.2]. The authors did not mention the cleaning or treatment process of the substrate used. Please clarify this information for the readers.

Ref.1 MRS Advances 8 (5), 194-200, 2023. https://doi.org/10.1557/s43580-023-00515-3

(b) The scale and numerical values in AFM are not legible. Consider the same comments for all relevant images in the manuscript.

(c) In lines#218-226, the authors discovered the substrate surface wetting phenomena by the coating solutions is the crucial factor for coherent and compact layer synthesis. Did the author measure contact angle of the coating solutions/solvents with the substrates used? How author choose suitable solvent(s) for compact layer by spin-coating?

(d) In lines#252-254, the authors wrote, “A tightly packed monolayer of hcp arrays can be detected in this case. The QDs in this layer were highly ordered and monodispersed in their size and shape. However, some amount of surface cracking was identified in the layer.” What did author mean by “hcp arrays”? What was the deposition/processing temperature of thin layer(s)?

(e) Clarify in the Fig. S6. Caption whether images are from S(TEM) images or FE-SEM images.

(f) Correct the Fig. 8 caption, Fig.8c.

(g) Can author include more magnified area-images of PbTe/TiO2 and TiO2/ITO interfaces with SAED for readers.

(h) In lines#430-433, the authors wrote, “The polarity and hydrophobicity of the surface highly influence the fabricated layer. PbTe QDs are known to be hydrophobic,

and for this reason, they create a better layer on hydrophobic TiO2/ITO glass substrate compared to hydrophilic ITO.[56]” The authors are advised to include contact angle of coating solutions/solvents with the substrates of interest to justify this claim.

5. Conclusion: Please re-write the conclusion part again based on the changes that would be made to the current manuscript and highlight the most important factor(s) that play critical role to attain compact and coherent monolayer of PbTe QDs.

Reviewer #2: Manuscript #: PONE-D-24-40936

Title: PbTe Quantum Dots Highly Packed Monolayer Fabrication by a Spin Coating Method

PLOS ONE

Dear Prof. Editor-in-Chief;

The mentioned manuscript subjects to corrections.

1. Polish the English language

2. To improve the manuscript quality, cite for QDs the followings; Quantum Dots: Synthesis, characterization, and optical investigations, IOP Publishing, Bristol, UK. © 2024; Graphene, Nanotubes and Quantum Dots-Based Nanotechnology: Fundamentals and Applications, Elsevier Inc., Netherlands. © 2022; Innovation Discovery 1 (2024) 18-25; Journal of Materials Science: Materials in Electronics 34 (2023) 993-1016; Luminescence 33 (2018) 260-266; Superlattices and Microstructures 88 (2015) 662-667; Canadian Journal of Physics 93 (2015) 1490-1494; Journal of Nanoelectronics and Optoelectronics 10 (2015) 705–710; Materials Science in Semiconductor Processing 39 (2015) 276–282; Solar Energy 115 (2015) 33–39; Renewable Energy 45 (2012) 232-236; Solar Energy 85 (2011) 2283-2287; “Quantum dots-based solar cells for potential application”, in: Mingheng Li (Editor), Energy Systems and Processes: Recent Advances in Design and Control, AIP Publishing, USA. © 2023; “Optical properties of quantum dots”, in: Y. Al-Douri (Editor), Graphene, Nanotubes and Quantum Dots-Based Nanotechnology: Fundamentals and Applications, Elsevier Inc., Netherlands. © 2022

3. Section 2 needs more elaborations

4. Tables 1-2. Any other data for comparison?

5. spherical QDs (6-9 nm) over cubical QDs (10-13 nm). How did you measure?

6. layer with the height of single QD) covering approximately 3 cm². How did you measure

7. Fig. 1. Highlight the thickness

8. Fig. 2. Highlight the grain size

9. QDs monolayer. How did you measure?

10. XRD is required to provide table and structural properties and crystallite size

11. Compare crystallite and grain sizes

12. What’s about optical studies?

13. Pb is a toxic. What’s the safety arrangement to do the experiment?

14. Highlight the advantages of spin coating method

15. Section 4 should reflect abstract

16. 34 of 62 refs is out of date (more than 10 years). Update!

6. PLOS authors have the option to publish the peer review history of their article (what does this mean?). If published, this will include your full peer review and any attached files.

Reviewer #1: **Yes: **Syed Farid Uddin Farhad

Reviewer #2: No

---

## [Author Response · Author response to Decision Letter 0]

25 Nov 2024

We greatly appreciate your input and valuable suggestions, which have significantly contributed to improving the quality of the manuscript. We have addressed all your comments and revised the manuscript to align with the journal's style requirements. Additionally, we have provided the relevant raw data to support our findings. 

Our point-by-point response is provided is separate pdf file.

---

## [Decision Letter · Decision Letter 1]

23 Dec 2024

PONE-D-24-40936R1PbTe Quantum Dots Highly Packed Monolayer Fabrication by a Spin Coating MethodPLOS ONE

Dear Dr. Minnes,

Thank you for submitting your manuscript to PLOS ONE. After careful consideration, we feel that it has merit but does not fully meet PLOS ONE’s publication criteria as it currently stands. Therefore, we invite you to submit a revised version of the manuscript that addresses the points raised during the review process.

We look forward to receiving your revised manuscript.

Kind regards,

María Eugenia Pérez Barthaburu

Academic Editor

PLOS ONE

Journal Requirements:

Additional Editor Comments:

Thank you for your work on this new version of your manuscript.

Before it can be accepted for publication, I suggest addressing the following points:

1. General Revisions:

Please revise the entire manuscript for grammatical accuracy and clarity.

2. Experimental Details:

Mention how you obtained square QDs. This information is missing both in the experimental description and in Table 1.

3. Specific Edits:

Line 155: Replace the term "purification" with "further cleaning."

4. Ligand Exchange:

In Section 2.4, specify which ligand was exchanged by PDA.

Detail the conditions (e.g., temperature) used during the exchange process.

Explain how you verified that the ligand exchange was successful.

5. Terminology Consistency:

In the abstract, it is stated that round QDs yield better results than cubic ones. However, Section 3 mentions cubic and cuboctahedral QDs. Ensure consistent terminology throughout the manuscript (e.g., rounded/spherical QDs).

6. Structural Adjustments:

The first paragraph of Section 3 (Results and Discussion) should be merged with the Introduction for better flow and coherence.

7. Citations:

Review the citations suggested by Reviewer 2. Retain only the strictly necessary ones, as nearly 10% of the citations are to Al-Douri Y.

8. Figures and Data Presentation:

The images in Figure S1 are unclear. They appear to be in STEM mode within an SEM setup rather than traditional SEM mode. Please clarify and correct this.

Correct the sequence of Figures S4 and S5 in the supplementary information. Since layers of 8.6 nm are mentioned earlier in the manuscript, Figure S5 should precede Figure S4.

Provide details on how the coverage percentage was measured.

**Comments to the Author**

1. If the authors have adequately addressed your comments raised in a previous round of review and you feel that this manuscript is now acceptable for publication, you may indicate that here to bypass the “Comments to the Author” section, enter your conflict of interest statement in the “Confidential to Editor” section, and submit your "Accept" recommendation.

Reviewer #2: All comments have been addressed

6. Review Comments to the Author

Reviewer #2: Correct Ref. [6] Journal of Materials Science: Materials in Electronics 34 (2023) 993-1016; Ref. [9] Y. Al-Douri “Quantum dots-based solar cells for potential application”, in: Mingheng Li (Editor), Energy Systems and Processes: Recent Advances in Design and Control, AIP Publishing, USA. © 2023, pp. 6-1–6-18; Ref. [10] Y. Al-Douri (Editor), Graphene, Nanotubes and Quantum Dots-Based Nanotechnology: Fundamentals and Applications, Elsevier Inc., Netherlands. © 2022; Ref. [15] Y. Al-Douri (Author), Quantum Dots: Synthesis, characterization, and optical investigations, IOP Publishing, Bristol, UK. © 2024; Ref. [26] correct: Renewable Energy

7. PLOS authors have the option to publish the peer review history of their article (what does this mean?). If published, this will include your full peer review and any attached files.

Reviewer #2: No

---

## [Author Response · Author response to Decision Letter 1]

30 Dec 2024

Respond to Editor is added in a separate file

---

## [Editor Report · Decision Letter 2]

3 Jan 2025

PbTe Quantum Dots Highly Packed Monolayer Fabrication by a Spin Coating Method

PONE-D-24-40936R2

Dear Dr. Minnes,

We’re pleased to inform you that your manuscript has been judged scientifically suitable for publication and will be formally accepted for publication once it meets all outstanding technical requirements.

Kind regards,

María Eugenia Pérez Barthaburu

Academic Editor

PLOS ONE
---

## [Editor Report · Acceptance letter]

8 Jan 2025

PONE-D-24-40936R2 

PLOS ONE

Dear Dr. Minnes, 

I'm pleased to inform you that your manuscript has been deemed suitable for publication in PLOS ONE. Congratulations! Your manuscript is now being handed over to our production team.

Kind regards, 

on behalf of

Dr. María Eugenia Pérez Barthaburu 

Academic Editor

PLOS ONE